# Assessment of dispersion metrics for estimating single-cell transcriptional variability

**Tina Chen** [1,2], **Laurie A. Boyer** [2,3]*, **Divyansh Agarwal** [2,4]*

1 Massachusetts Institute of Technology, Department of Electrical Engineering and Computer Science, Cambridge, Massachusetts, United States of America, 2 Massachusetts Institute of Technology, Department of Biology, Cambridge, Massachusetts, United States of America, 3 Massachusetts Institute of Technology, Department of Biological Engineering, Cambridge, Massachusetts, United States of America, 4 Massachusetts General Hospital, Department of Surgery, Boston, Massachusetts, United States of America

* lboyer@mit.edu (LAB); agarwald@mit.edu (DA)

## Abstract

Single-cell RNA sequencing data enables analysis of transcript levels of single cells across different cell types and conditions. Recent work has highlighted the value of measuring gene-specific transcriptional variability, or noise, within a genetically identical population of cells in addition to mean expression, given that these differences contribute to biological processes including development and disease. However, measuring transcriptional noise remains a challenge. Here, we systematically compared statistical methods by simulating single-cell data by varying both dispersion and count size to assess the relative responsiveness to noise of several commonly used statistical metrics: the Gini index, variance-to-mean ratio, variance, coefficient of variance (CV), $CV^2$, and Shannon entropy. We found that the variance-to-mean ratio scales approximately linearly with increasing dispersion and is independent of dataset size. In contrast, the Gini index displayed paradoxical behavior in that it increases as dispersion decreases, and Shannon entropy was not scale-invariant. Next, we applied the variance-to-mean ratio (Fano factor) to measure transcriptional variability in single-cell datasets representing different complex systems and cross-platform measurements. Our data show that many genes display transcriptional variability within the same cell type, and that while variation does not correlate with gene characteristics such as transcript level, promoter GC content, or evolutionary gene age, variable genes are often correlated with specific biological processes. Notably, most genes and pathways with highest transcriptional variability as identified by the Fano factor were largely independent of differentially expressed genes and have also been implicated in biological processes related to the system. Thus, our data highlight that choice and application of appropriate models for measuring transcriptional variation in scRNA-seq data can reveal biologically relevant information beyond what is observed from mean expression alone.

**Data availability statement:** UMI counts from Manivannan et al. are available at GEO with accession number GSE193746. UMI counts from Lana-Elola et al. (10.1126/scitranslmed.add6883) are available at GEO with accession number GSE196447. UMI counts from Yuzwa et al. (10.1016/j.celrep.2017.12.017) are available at GEO with accession number GSE107122. The code used to generate the figures in this manuscript are available on Github: https://github.com/lboyerlab/Comparison_of_dispersion_metrics/tree/main.

**Funding:** This work was supported by the National Heart, Lung, and Blood Institute (R01HL140471 to LAB). The funder had no role in study design, data collection, and interpretation, or the decision to submit the work for publication.

**Competing interests:** The authors have declared that no competing interests exist.

## Author summary

Single-cell RNA sequencing (scRNA-seq) data allows for the study of transcriptional variability. However, the contribution of transcriptional variability to gene expression has not been fully appreciated in part due to a lack of consensus on how to estimate and apply noise metrics for downstream analytical modeling. The study of transcriptional variability provides a new lens through which we can study how transcriptional dynamics impact complex biological phenomena. Here, we simulated single-cell data to test six dispersion metrics for their relative sensitivity to variability in single-cell counts. From our simulations, we found that the variance-to-mean ratio (VMR or Fano factor) appears to be the most suitable metric among those tested for quantifying transcriptional variability as it is scale-invariant and is easily interpretable with respect to changes in data dispersion. We then applied the VMR to analyze changes in transcriptional variability in scRNA-seq datasets from platforms with different capture rates. We find that the Fano factor can identify genes distinct from differentially expressed genes and that variable genes relate to specific functional categories that likely reflect the underlying biology. For most distributions, VMR/Fano factor is a reasonable, robust choice for modeling transcriptional noise. However, for certain niche distributions, other metrics may be better suited. Together, we demonstrate that model choice for measuring transcriptional variability can provide new biological insights into how cells respond and adapt in complex systems.

## Introduction

Single-cell RNA sequencing (scRNA-seq) approaches have enabled the study of gene expression in individual cells to better appreciate stochastic biological processes and precisely map cell states. Despite its evolution over the last decade, scRNA-seq data remain sparse in any given experiment, with a large proportion of dropouts, which occur when transcripts are not captured due to efficiency and sequencing depth challenges. Nevertheless, capturing transcriptional information at single-cell resolution provides a unique opportunity to study gene transcriptional variability within a genetically identical population of cells. Investigating transcriptional noise can reveal critical features of cell state dynamics and infer gene regulatory principles; however, differentiating biological noise from technical noise in scRNA-seq data remains a challenge.

Transcriptional variability, also referred to as transcriptional noise, biological noise, or single-cell expression variability, has both stochastic and deterministic components [1]. This noise, thought to affect cell fate and function, inevitably arises in biological processes due to the kinetics of reversible binding and random collisions between proteins and nucleic acids [2]. A number of factors contribute to variability including transcriptional bursting [3,4] promoter architecture [5,6], and chromatin modifications [6,7]. For example, higher numbers of alternate transcriptional start sites and the presence of a TATA box are correlated with high transcriptional variability, whereas

genes associated with CpG islands tend to display low transcriptional variability [6]. Thus, transcription dynamics could convey critical information about how gene programs are regulated.

Although the role of transcriptional variability remains controversial, it appears to play key roles in several biological processes including development, disease, and aging [8]. For instance, increased transcriptional variability is associated with a cell state prior to exit from pluripotency during embryonic development [9] and with the robustness of the immune response in mouse dendritic cells [10]. Transcriptional variability has also been implicated in the underlying differences in tumor heterogeneity [11], as well as in aging, where epigenetic drift can impact precise control over gene expression programs. For example, transcriptional noise was found to be higher in cardiomyocytes of old mice [12] and in aged human pancreas [13].

More generally, the "variation is function" hypothesis proposes that transcriptional variability could be important for higher function in multicellular organisms [14,15]. Another perspective on transcriptional variability comes from information theory. Modeling the kinetics of reversible binding of transcription factors (TFs) suggests that transcriptional variability is necessary for the transmission of information, increases pathway sensitivity to signal variation, and broadens the range of the response distribution [16]. Despite the evidence, lack of consensus on how to measure transcriptional noise limits our ability to understand how these properties contribute to biology and disease. Current approaches range from classical statistical methods, such as the coefficient of variation (CV) and variance-to-mean-ratio (VMR; also known as the Fano factor) [12], to newer approaches like distance-to-centroid metrics and regression approaches [13,17]. Notably, analysis of transcriptional variability in the same aging datasets yielded conflicting results when different measures of transcriptional noise were used [18], highlighting the need for an improved understanding of how various dispersion metrics perform.

Other factors that impact model choice include the statistical and mechanical assumptions of each model. For example, the beta-Poisson distribution has been shown to arise from the telegraph model of two-state gene transcription at steady state [19]. On the other hand, the negative binomial distribution captures transcriptional bursts under the telegraph model [20]. Attention to non-normality in gene expression has been previously discussed and implicated in several oncologic pathophysiologic processes [21]. For instance, a bimodal distribution has been surmised to model genes with distinct on and off transcriptional states, and these switch-like genes have been shown to identify patient subgroups with different survival rates [22,23]. Thus, we sought to systematically compare different noise metrics across a range of distributions that capture various possible scenarios to model the underlying transcriptional output.

Analysis of transcriptional variability is further complicated by the limitations and characteristics of scRNA-seq and by technical noise. Mean expression and variance scale nonlinearly in scRNA-seq data, posing an analytical challenge. scRNA-seq data also have a high proportion of zeros due to both technical limitations and low expression levels. Normalization techniques, which range from scaling by sequencing depth to modeling the count-depth relationship for each gene within a cell [24], can reduce, but not eliminate, the effect of technical noise. Thus, it is important to differentiate between the transcriptional variability captured by scRNA-seq and molecular and phenotypic variability. While existing work has used gene-level variance to identify variable genes, this utility is often limited to dimensionality reduction and relies on using a predetermined variance metric. The goal of our current work was to systematically test the robustness of six commonly used statistical methods for quantifying transcriptional variability in single-cell data across a range of theoretical and empirical scenarios. We observe that the VMR, commonly referred to as the Fano factor, is a robust measure of transcriptional variability in single-cell counts and can be used to identify biologically relevant genes and pathways not found in differential expression analysis in existing scRNA-seq datasets. Our work and analysis pipelines can be broadly applied to studying the role of transcriptional variability in dynamic processes such as development and disease.

## Results

### Simulations assess relative sensitivity to variability in single-cell counts

We first compared commonly used dispersion metrics for the quantification of transcriptional variability. The Gini index, which is often used to measure economic inequality; the VMR or Fano factor, a measure of deviation from the Poisson

distribution; Shannon entropy, a measure of uncertainty in information; coefficient of variation (CV), a measure of standardized dispersion; and squared coefficient of variation ($CV^2$) are metrics that have been used for quantifying transcriptional noise [17,25] and clustering [26,27]. Sample variance was also included as a simple and intuitive measure of variability. Empirically, these metrics can underestimate heavy-tailed and over-dispersed distributions; therefore, our simulations sought to better uncover any unintuitive behavior that may not be evident from a closed-form expression and more directly compare the metrics to each other based on the underlying data distribution.

To assess the sensitivity of the six metrics to variability in scRNA-seq data, we simulated scRNA-seq data using several distributions, including the Poisson, Poisson-lognormal, negative binomial (or gamma-Poisson), beta-Poisson, hurdle negative binomial (hurdle-NB), and uniform distributions. The negative binomial and beta-Poisson distributions were chosen as models that represent scRNA-seq data [28,29], and the parameters for each distribution were chosen to approximate scRNA-seq data, as described in the Methods. For each distribution, we generated multiple instances of the same distribution with differing levels of dispersion (examples of common distributions are shown in Fig 1A). From each instance, we simulated counts matrices of varying sizes. Counts for each gene within a given counts matrix were independently simulated across cells. All genes within a given counts matrix were simulated from the same distribution, with the same parameterization and dispersion. More specifically, for each family of distributions (i.e., Poisson, negative binomial, etc.), we generated 100 instances of the distribution with increasing levels of dispersion. For each instance of a distribution, we simulated 50 counts matrices, which all had the same number of genes but varying numbers of cells. All genes within a given matrix were simulated from the same distribution instance (i.e., same level of dispersion) and assuming no gene-gene correlation. We first calculated the expression

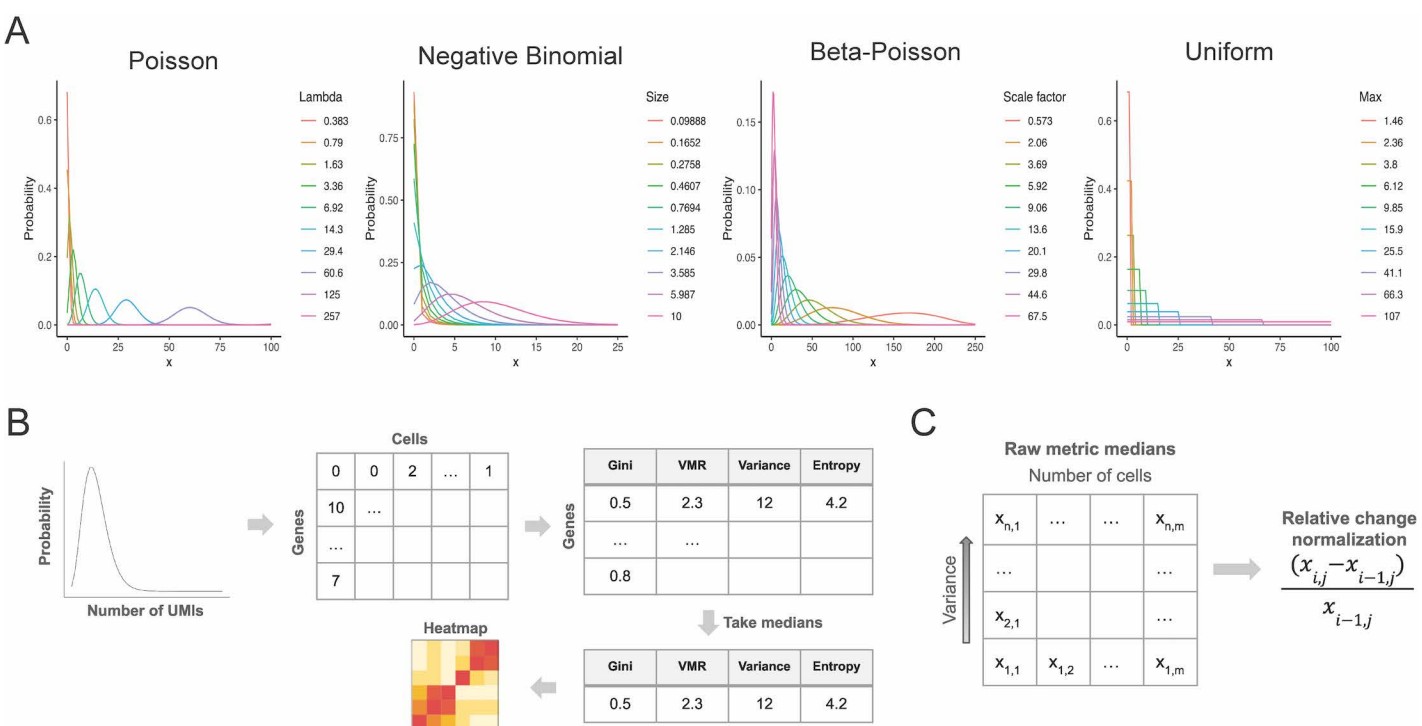

**Fig 1. Simulations of single-cell data reveal relative sensitivity of dispersion metrics across various data distributions. (A)** Examples of probability density functions of the Poisson, negative binomial, beta-Poisson, and uniform distributions with varying levels of dispersion, which were used to generate simulated single-cell counts. **(B)** Schematic of simulations of single-cell data. The Gini index, VMR, variance, entropy, CV, and $CV^2$ were applied to the simulated data to assess their sensitivity to variability in the underlying distribution. The results were visualized in heatmaps. **(C)** Schematic for normalization of dispersion metrics.

variability from the simulated counts using the metrics described above. Next, the metrics were normalized to assess their relative sensitivity to expression variability (Fig 1B and 1C). Heatmaps were generated from the raw and normalized metric values to visualize how the metrics scale with increasing dispersion and with increasing dataset size.

Because the dispersion in the underlying sampling distributions increased by a fixed exponential rate, a metric that scales perfectly with the theoretical dispersion would be expected to have identical normalized values with increasing dispersion. To assess this quality, the mean of the normalized, or relative change, values for each metric within each simulation was calculated, and the deviation of the normalized metric values from their respective means was obtained. The percentage of normalized metric values for each metric that falls within a standard distance (0.01, 0.005, or 0.0025) from the mean normalized value was calculated to quantify the spread of the normalized metric values.

Across the distributions, we observed the VMR/Fano factor, variance, CV, and $CV^2$ to be scale-invariant in terms of being independent of dataset size (Fig 2). In contrast, at the same level of dispersion across distributions, such as uniform and beta-Poisson, entropy values increased with the increasing size of the dataset (S1-S4 Figs). Scale invariance is an important property for quantifying transcriptional variability as transcriptional variability should be theoretically independent of dataset size. In general, the VMR, variance, and entropy increased as the dispersion in the sampling distributions increased, except in the uniform distribution, wherein entropy remained the same as the dispersion increased. Notably, the Gini index, CV, and $CV^2$ decreased as the dispersion in the sampling distributions increased. This behavior is paradoxical, as a metric of transcriptional variability is expected to scale with dispersion in the data. We also performed similar simulations where the number of genes in the matrices was altered rather than the number of cells and found similar results (S1-S4 Figs). Simulations to assess the impact of gene-gene correlation on metric performance yielded concordant results wherein the Fano factor, CV, and $CV^2$ emerged as robust estimators of transcriptional noise across a range of gene-gene correlations (S5 Fig).

These simulations suggest that the Gini index and entropy are less reliable metrics for measuring transcriptional variability as entropy depends in part on the size of the dataset and the Gini index exhibits paradoxical behavior. In our simulations, variance and the VMR scaled approximately linearly with increasing dispersion, independent of dataset size. However, because variation increases in proportion to gene expression and noise metrics are focused on modeling variability rather than mean expression, VMR, which captures relative variability, is more suitable for quantifying transcriptional variability than the variance, which captures absolute variability. Thus, among the statistical metrics tested, the VMR/Fano factor appears to be a robust metric for quantifying transcriptional variability.

Additionally, we simulated scenarios wherein the UMI counts may follow an overdispersed model using a Poisson-lognormal distribution, as described in the Methods. As the theoretical dispersion increased, the Gini index, CV, and $CV^2$ decreased, and demonstrated the largest spread of relative change, implying that they may be less robust in this scenario compared to other metrics (S6 Fig). Shannon entropy appeared to scale with increasing dispersion in the underlying Poisson lognormal with the tightest spread of relative change values. Second to entropy, VMR/Fano Factor and variance also demonstrated tighter spread of noise estimates, suggesting that they may be more robust for overdispersed data compared to metrics such the Gini index, CV, and $CV^2$ (S6 Fig).

To further assess a common scRNA-seq feature– zero inflation– we also tested our metrics on hurdle-NB distributions. We simulated counts from a hurdle-NB distribution as described in the Methods. We observed that while all the metrics scaled well with the theoretical variance, entropy and Gini index were more robust and displayed a tighter distribution of noise estimates when the proportion of zeroes was 90% (hurdle-NB probability = 0.1) and became less robust as the proportion of zeroes decreased. On the other hand, Fano factor, CV and $CV^2$ appeared to be relatively stable in both hurdle-NB scenarios (S7 Fig), suggestive of their advantageous properties for zero-inflated data.

## The Gini index decreases as variability in the simulated scRNA-seq counts increases

Recently described paradoxical behavior of the Gini index in the negative binomial distribution [30] was also observed across the sampling distributions tested in our simulations (Fig 2). As the theoretical variance of the sampling distributions

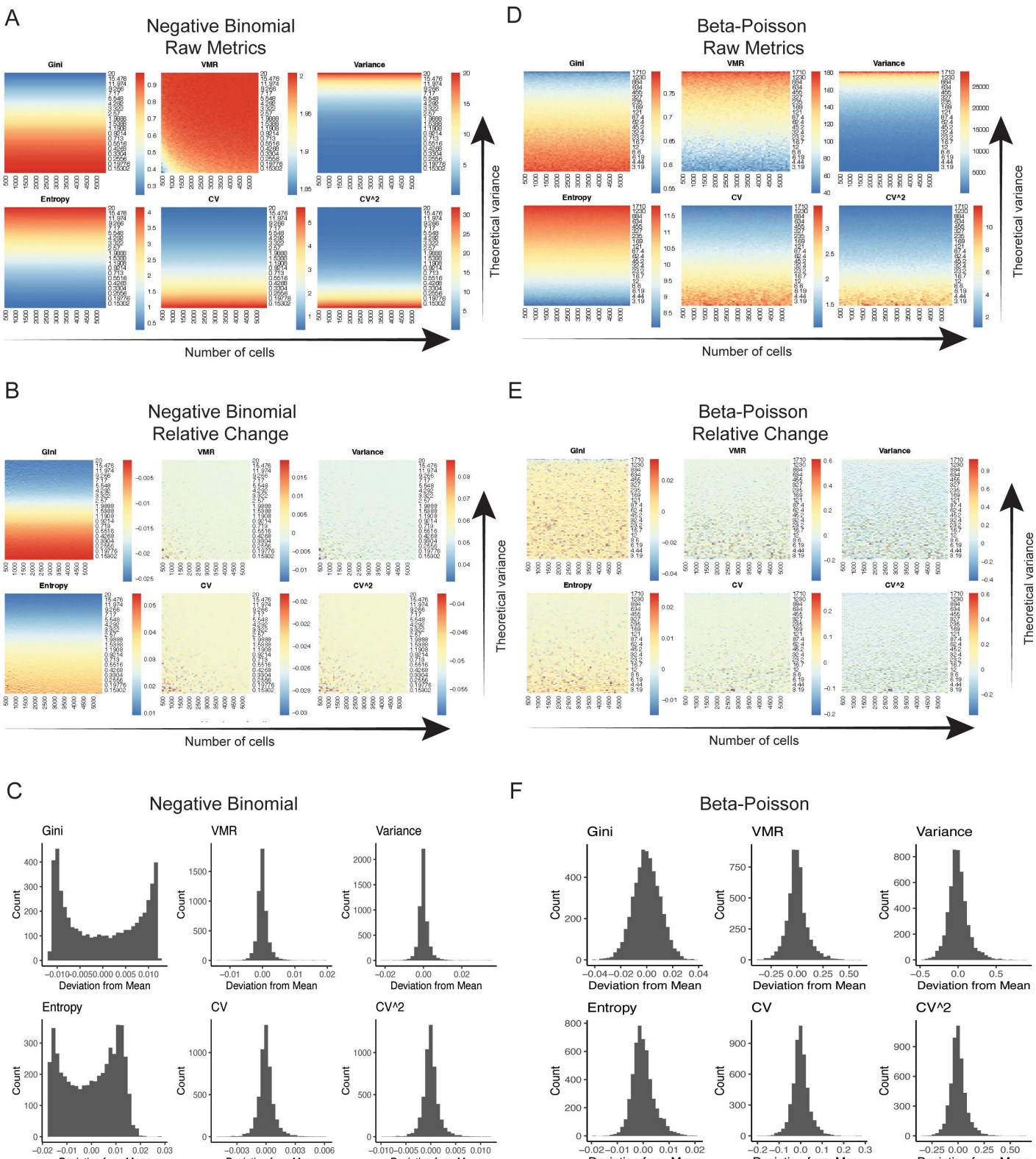

**Fig 2. VMR is most sensitive to changes in dispersion. (A)** Heatmaps of each metric applied to simulated counts drawn from instances of the negative binomial distribution. The dispersion in the sampling distributions increases across the y-axis, as determined by the size parameter r for the

negative binomial. The size of the simulated data increases across the x-axis, as determined by the number of cells in each counts matrix. **(B)** Heatmaps of relative change in each metric applied to counts from **(A)**. **(C)** Histograms of deviation of relative change from mean relative change for each metric as calculated from the distributions described in **(A)**. **(D - F)** Same as above for counts simulated from the beta-Poisson distribution. The theoretical variance increases as determined by a scaling factor on the β parameter.

increased, the Gini index decreased. An intuition for this paradoxical behavior can be observed from the shapes of assessed distributions (Fig 3A). For example, in the negative binomial distribution, the Gini index indicates highest noise when the mean and variance of the distribution are less than 1, which corresponds to data with a high proportion of zeros. When the mean and variance of the negative binomial are greater than 1, both the data and the dispersion approximate

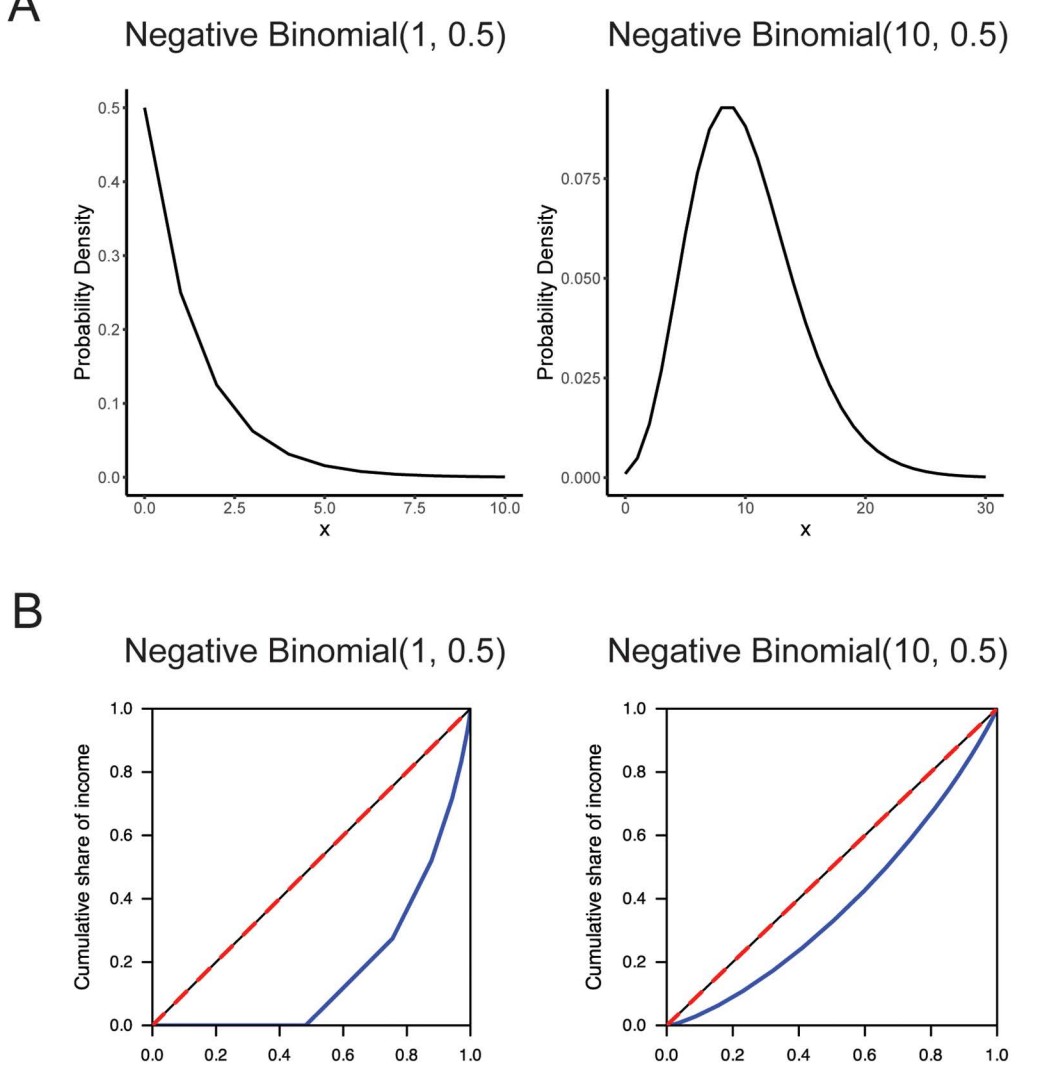

**Fig 3. Examining the paradoxical behavior of the Gini index. (A)** The probability density functions of negative binomials with $r = 1$ and $r = 10$, both with $p = 0.5$. **(B)** Lorenz curve (blue) of data sampled from a negative binomial with $r = 1$ and $r = 10$, both with $p = 0.5$. The line of equality is shown in red.

that of a normal distribution, and the Gini index indicates lower noise. Conversely, when the mean and variance are less than 1, the sample variance of the data is low whereas the sample variance is high when mean and variance are greater than 1. These two cases illustrate the paradoxical relationship between the Gini index and metrics like variance, which can be understood from two perspectives.

One explanation of this behavior comes from the context of the use of the Gini index in measuring economic inequality in a population. First, consider population A, where half of the individuals have $0, and the other half has between $1 and $3. Because half of the population has no money, we would say this is a population with high economic inequality, since the relative disparities between individuals are high. Next, consider population B, where on average people have $10 and money is approximately normally distributed with a standard deviation of $4.5 (in other words, a negative binomial with probability 0.5 and $r$ of 10, where $r$ is the target number of successes). Intuitively, population B is less economically disparate. Although the range in B is larger than the range in A, the relative differences between the amount of money individuals have are lower in B than in A.

Mathematically, the Gini index is derived from a Lorenz curve [31], which plots the cumulative income in a population against the cumulative number of people in the population ranked from lowest to highest income. The Gini index is computed as the ratio of the area between the Lorenz curve and the line of equality to the total area under the line of equality. Therefore, the Lorenz curve of a negative binomial distribution with small $r$ and a large proportion of zeros would have very low cumulative income over the first half of the graph, and then a sharp increase in cumulative income as the cumulative population nears the total population (Fig 3B). This shape yields a large area between the line of equality and the Lorenz curve. On the other hand, the Lorenz curve of a population drawn from a negative binomial distribution where $r$ is greater than 1 more closely resembles the line of equality and produces a smaller area and smaller Gini index.

These examples, and the paradoxical behavior of the Gini index, compel careful consideration of when this metric should be used as a proxy for biological noise. In biological terms, is a gene noisier when it is expressed in some cells but not in others, or when it is expressed in all cells but at varying levels? This dichotomy has also been described as digital (on/off) versus analog (continuous) noise [10]. Transcriptional variability does not conform to the conventional notion of dispersion, explaining at least in part why the Gini index decreases as the variance of the negative binomial distribution increases. Whether the behavior of interest is best modeled by the spread of a distribution or the relative disparity within a distribution should inform whether the Gini index or metrics such as the VMR are more appropriate.

Based on our application of these metrics to scRNA-seq data, we find that depending on the features and properties of the genes of interest, different measures of statistical heterogeneity can find applications as a proxy for transcriptional noise. In some instances, the Gini index may be suitable for capturing genes with specific kinetic properties such as genes that switch on or off rapidly. For example, the Gini index separated pluripotency factors whose expression drops off quickly versus more gradually during differentiation [32]. However, in general, the Fano factor may be a more robust metric of transcriptional variability that scales approximately linearly with increasing dispersion. Further still, for certain overdispersed or niche data distributions, CV and $CV^2$ might be good choices as well.

## Association of transcriptional variability with gene characteristics

A number of gene characteristics are involved in transcriptional variability, including transcriptional burst frequency and elements of gene architecture such as the number and nature of regulatory elements. However, neither a complete list of factors that determine transcriptional variability nor their individual contributions to variability have been fully elucidated. Therefore, we tested whether gene characteristics such as transcript levels and gene length are associated with changes in transcriptional variability, as measured by the VMR.

To this end, we first applied the VMR to quantify transcriptional variability in a publicly available scRNA-seq dataset that analyzes the impact of maternal hyperglycemia (matHG) on congenital heart disease (CHD) using embryonic mouse hearts at E9.5 and at E11.5 [33]. This dataset was chosen for its 10x Genomics sequencing platform given its availability

and widespread applicability. We also chose this dataset for its relatively deep sequencing with an average read depth of at least 50,000 reads per cell, which further limits any technical noise introduced by shallow sequencing. We quantified gene-specific transcriptional variability at E9.5 and E11.5 in cardiomyocytes for both control and matHG conditions. We then calculated the difference in transcriptional variability of each gene between the conditions within each timepoint. Using the difference in VMR between the conditions, we identified genes with high and low transcriptional variability across cardiomyocytes (S1 Table). Lowly expressed genes were excluded from analysis to minimize technical noise.

Next, we examined the correlation between absolute change in transcriptional variability at each time point and four gene characteristics implicated in noise: transcript level, gene length, promoter GC content, and evolutionary gene age. Evolutionary gene age was examined because studies suggest older genes have less transcriptional variability than younger genes [34]. However, much remains unknown about the role of transcriptional variability in evolution and how a balance between maintaining essential functions and facilitating evolution can be struck [35]. We found a paucity of correlation between absolute change in transcriptional variability and the gene-level factors examined (Fig 4). Absolute changes in transcriptional variability and transcript levels were weakly negatively correlated at E9.5 ($\tau = -0.19$) and weakly positively correlated at E11.5 ($\tau = 0.13$). Absolute changes in transcriptional variability and gene length showed a weak correlation approaching 0 at E9.5 ($\tau = 0.020$) and at E11.5 ($\tau = -0.060$). Additionally, the direction of $\tau$ depends on the threshold of mean normalized expression that was used to filter out lowly expressed genes (Fig 4B). Absolute change in transcriptional variability and promoter GC content demonstrated no substantial correlation at E9.5 and very weak correlation at

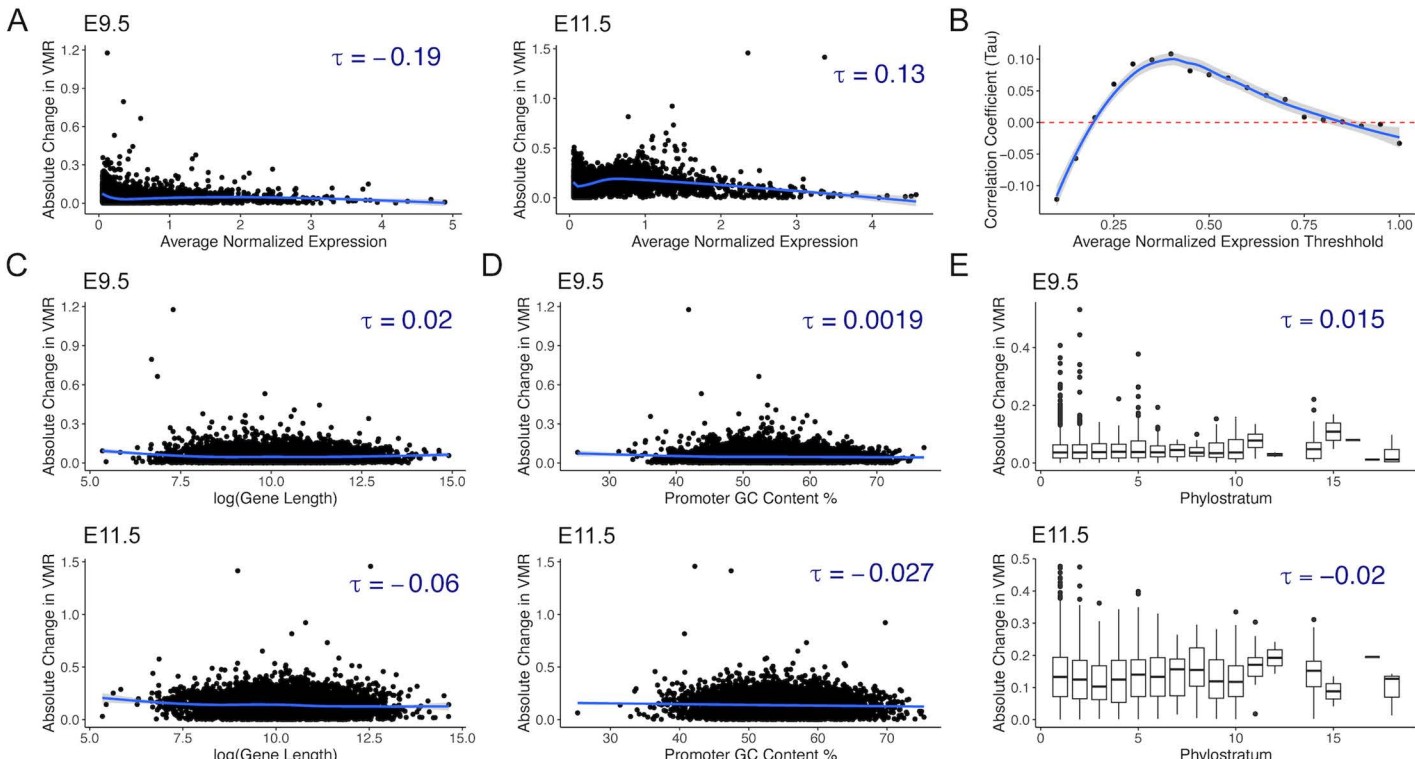

**Fig 4. Absolute change in transcriptional variability is not correlated with transcription level, gene length, promoter GC content, and evolutionary gene age. (A)** Absolute change in transcriptional variability at E9.5 and at E11.5 is extremely weakly correlated with transcript levels as measured by average normalized counts. **(B)** The correlation coefficient of absolute change in transcriptional variability with transcript level varies with the threshold for removing lowly expressed genes at E9.5. Absolute change in transcriptional variability is also extremely weakly correlated with gene length **(C)**, promoter GC content **(D)**, and gene phylostrata **(E)**.

E11.5 ($\tau$=-0.027) (Fig 4D). There was no significant correlation between absolute change in transcriptional variability and gene phylostrata, and the magnitude of $\tau$ was <0.02 at both time points in this dataset.

To determine whether our findings were unique to this dataset, we performed the same analysis in 830 cardiomyocytes from a mouse model of trisomy 21 (T21) [36]. Although the genes with highest variability in T21 cardiomyocytes were largely different from those with highest variability in the matHG cardiomyocytes, we observed a similarly weak correlation between absolute change in transcriptional variability and transcript levels ($\tau$=-0.094), promoter GC content ($\tau$=-0.034), and gene phylostrata ($\tau$=0.027) (S8 Fig). The correlation between absolute change in transcriptional variability and gene length was also weak, and not statistically significant ($\tau$=-0.012). These results suggest that measuring transcriptional variability captures an aspect of gene expression that is relatively orthogonal to, and independent from, characteristics such as transcript level, gene length, promoter GC content, and evolutionary age.

## Transcriptional variability reveals biological insights distinct from differential gene expression

To further assess the use of VMR/Fano Factor for quantification of transcriptional variability, we performed further analysis of the matHG dataset. First, we asked if genes with large changes in transcriptional variability between conditions over-lapped with differentially expressed genes (DEGs), which are the genes most commonly reported in data analysis. Again, lowly expressed genes were excluded from analysis to minimize technical noise. We found that the 100 most significant DEGs showed little overlap with the top 100 genes with the largest absolute change in VMR (<4.2% overlap) at both E9.5 and E11.5 compared to controls (Fig 5A and S1 Table). When we increased our comparison to the 1000 most significant DEGs against the 1000 genes with largest change in VMR, we similarly found little overlap at both time points (<6% overlap). These data suggest that change in VMR can be used to identify genes that may be biologically relevant but not necessarily exhibit a change in mean differential expression.

To investigate the functional differences between DEGs and genes with large changes in transcriptional variability, we next performed gene set enrichment analysis (GSEA) using the 1000 most significant DEGs and compared the results to using the 1000 genes with the largest absolute change in VMR. For GSEA using DEGs, we used log-fold change to rank genes, and for GSEA using transcriptional variability, we used the change in VMR. Notably, Kyoto Encyclopedia of Genes and Genomes (KEGG) pathway analysis showed no overlap in enriched pathways between DEGs and genes with the largest changes in transcriptional variability (Fig 5B and S1 Table). For example, DEGs were enriched for pathways related to reactive oxygen species and disease states associated with hyperglycemia at E9.5, whereas signaling path-ways related to proliferation and growth were enriched in the transcriptional variability gene set.

Next, we asked if these differences could be used to identify potential upstream TFs. We performed TF motif enrich-ment analysis using the package RcisTarget [37] on sequences within 10 kbp of the transcription start sites of the 100 most significant DEGs and of the 100 genes with largest absolute change in VMR. We found that enriched TF motifs differed between the two groups (Fig 5C and S1 Table). For example, distinct motifs found among the genes with largest change in VMR include *Tead, Fos, Jun,* and *Ctcf*. Notably, *Tead, Fos*, and *Jun* are part of the Tead-AP1 axis regulated by Hippo signaling, a pathway identified in our GSEA of the transcriptional variability gene set that has also been impli-cated in the link between gestational diabetes and CHD [38]. Notably, many genes in the Hippo signaling pathway are not DEGs, suggesting that transcriptional variability analysis may capture information missed by differential expression (Fig 5D and S9 Fig). Candidate TFs and their predicted genes were then used to analyze downstream gene regulatory net-works (Fig 5E). Although the TFs enriched among the genes with large changes in transcriptional variability did not have notably high levels of noise, their corresponding target genes were all among the top 100 genes with largest absolute change in VMR. We also repeated the analysis described above on the T21 cardiomyocyte dataset [36]. Similarly, genes with large changes in transcriptional variability revealed distinct pathways compared to differential expression analysis (S2 Table), suggesting that quantifying transcriptional variability has potential as a tool for identifying genes and pathways involved in complex phenotypes that might otherwise not be obvious in conventional data analysis.

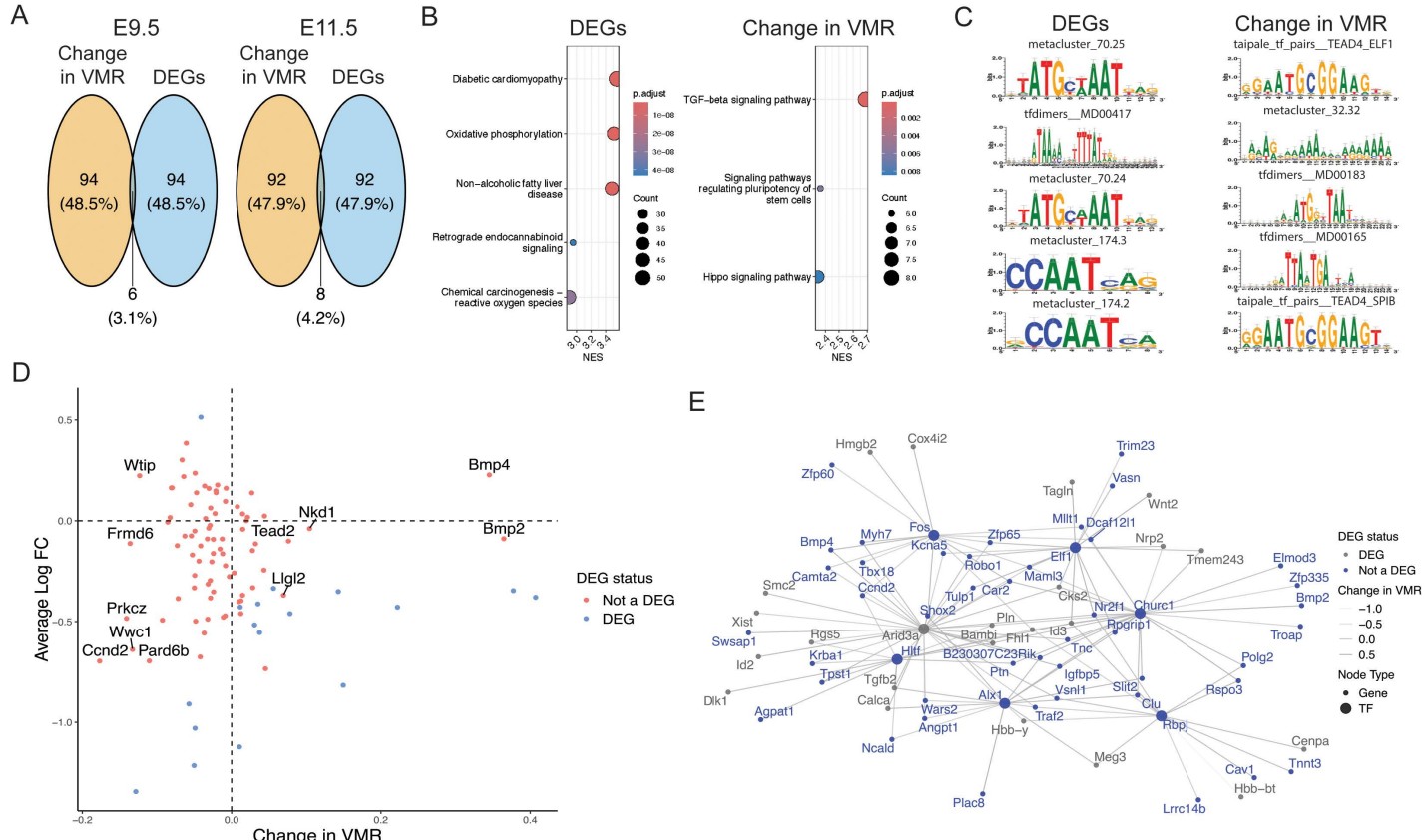

**Fig 5. Transcriptional variability confirms known pathways involved in maternal hyperglycemia and suggests new targets. (A)** Venn diagram of the 100 most significant DEGs and the 100 genes with largest absolute change in VMR at E9.5 and at E11.5. There were 760 CMs in the control E9.5 conditions, 827 CMs in the matHG E9.5 condition, 752 CMs in the control E11.5 condition, and 1163 CMs in the matHG E11.5 condition. The significance of a DEG is measured by its adjusted p-value from differential gene expression testing using the Wilcoxon rank sum test. **(B)** GSEA using KEGG pathways on the 1000 most significant DEGs, ranked by log-fold change, and on the 1000 genes with largest absolute change in VMR, ranked by change in VMR at E9.5. **(C)** TF motif enrichment was performed on the 100 most significant DEGs and on the 100 genes with largest absolute change in VMR, and the top 5 enriched motifs from each are shown [37]. **(D)** Change in VMR versus average log-fold change in expression for genes in the KEGG Hippo signaling pathway. DEGs at E9.5 are shown in pink. **(E)** Dendrogram of TFs enriched among the 100 genes with largest absolute change in VMR at E9.5 and their target genes. Opacity of edge weights is determined by the change in VMR of the genes. DEGs are shown in gray.

Furthermore, to determine whether estimates of transcriptional variability are robust across different platforms with inherent variations in technology, depth of coverage, and mean UMI count per cell, we applied VMR/Fano factor to estimate transcriptional variance in a non-10X dataset. We analyzed Drop-seq data from mouse neurons at E11.5 and E17.5 [39] and similarly found that utilizing change in Fano factor for GSEA and TF motif analysis identifies additional pathways and genes from those found by differential expression alone (S10 Fig).

Lastly, we examined the expression patterns of the genes showing changes in VMR. Specifically, we wondered whether the observed VMR changes reflect shifts in the fraction of expressing cells across the population, or instead correspond to 'analogue' changes in expression levels within individual cells. To address this point, we compared the observed changes in VMR to both change in fraction of non-zero UMI expressing cells and change in mean expression in three different datasets used previously [33,36,39] (Fig 6). We found a negative correlation between changes in VMR and both of these variables, suggesting that mean expression and fraction of non-zero UMI expressing cells indeed directly influence changes in VMR. However, the degree of correlation varies between different datasets and

sequencing platforms. Next, we examined the distribution of UMI counts between timepoints. In the murine neuron data, some genes, such as *Neurod6*, displayed a large shift in fraction of expressing cells. Others, like *Stmn2*, had a relatively smaller change in fraction of expressing cells, and instead displayed a change in the UMI distribution (Fig 6A-6C). *Neurod6*, a DEG, had the largest absolute change in VMR between timepoints, whereas *Stmn2*, a non-DEG, was among the top 20 genes with largest absolute change in VMR. In a10x Genomics dataset, change in VMR between cardiomyocytes in control and matHG conditions also had an inverse correlation with change in fraction of expressing cells and change

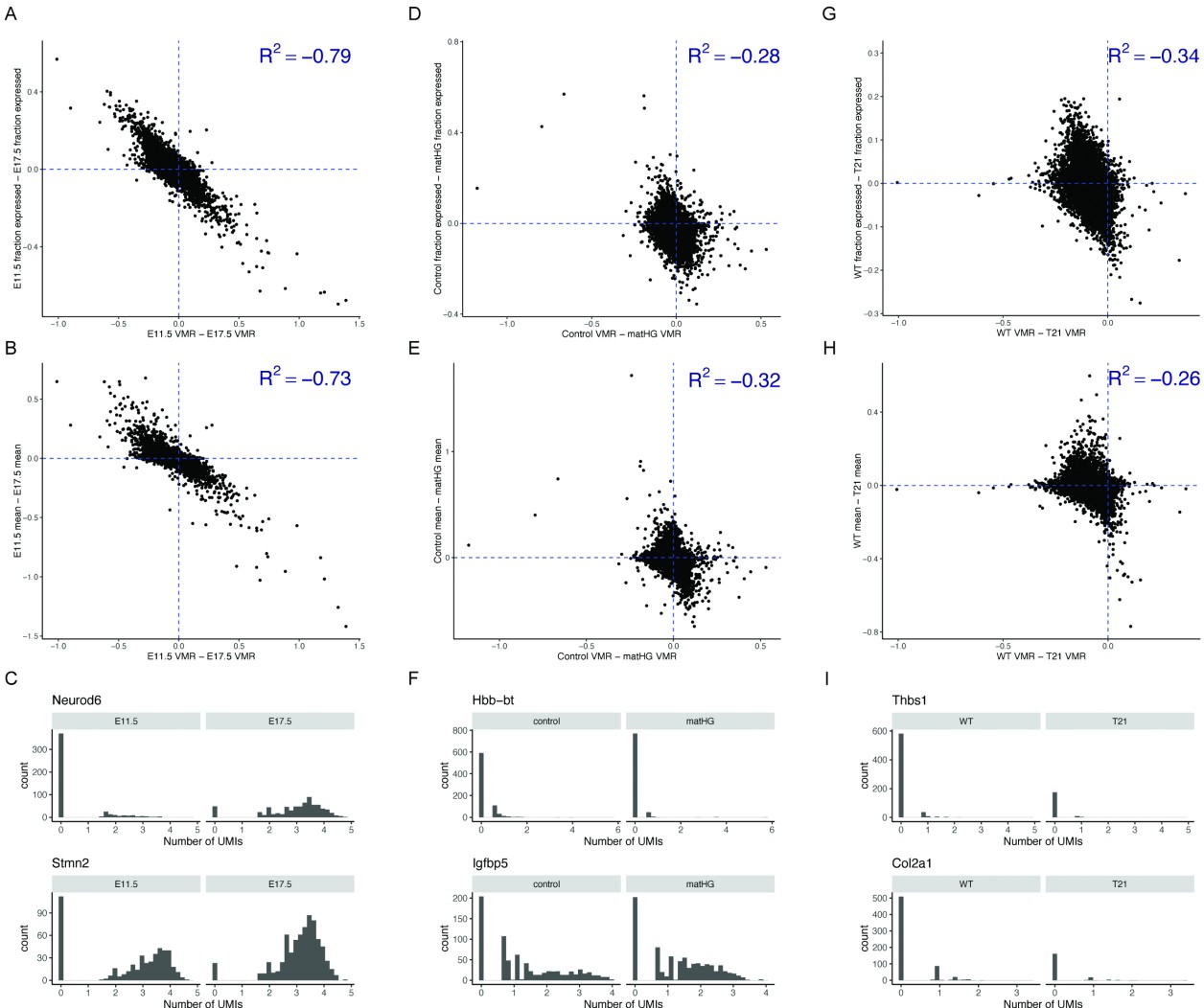

**Fig 6. Change in VMR has a negative correlation with mean and fraction of expressing cells and varies between platforms.** In a Drop-seq neuron dataset, change in VMR inversely correlates with (A) the fraction of expressing cells and (B) change in mean expression. **(C)** *Neurod6*, a DEG, is the gene in this dataset with the largest absolute change in VMR between E11.5 and E17.5, whereas *Stmn2*, a non-DEG, also had a large absolute change in VMR between timepoints. In the 10x Genomics cardiomyocyte dataset comparing control and matHG conditions at E9.5, associations between VMR and UMI counts across cells behave similarly **(D, E)**. Panel (F) shows *Hbb-bt*, a DEG with the largest absolute change in VMR at E9.5 between control and matHG. *Igfbp5* had the 10th largest absolute change in VMR between conditions, and is not a DEG. A negative correlation between VMR changes and UMI distribution is again evident in the 10x Genomics dataset comparing WT and T21 cardiomyocytes **(G, H)**. Panel (I) shows *Thbs1*, a gene with the largest absolute change in VMR between WT and T21 conditions. *Col2a1* had the 10th largest absolute change in VMR between conditions, and had the largest fraction of expressing cells among the top 10 genes with largest change in VMR. Neither *Thbs1* nor *Col2a1* are DEGs.

in mean expression (Fig 6D- 6F). Examination of the top genes with largest absolute change in VMR similarly found both genes that displayed a large change in fraction of expressing cells, like *Hbb-bt*, and genes that displayed a large change in shape of UMI count distribution, like *Igfbp5*. Of note, *Igfbp5* encodes insulin-like growth factor binding protein 5, which has been implicated in the progression of gestational diabetes mellitus in humans [40]. Similar analysis using a 10x Genomics dataset comparing WT and T21 cardiomyocytes in mice [36] found comparable results (Fig 6G- 6I). Collectively, our analyses suggest that transcriptional variability can be a useful tool for investigating underlying biological mechanisms irrespective of the sequencing technology.

## Discussion

Here we performed a comparison of the sensitivity and behavior of six metrics for quantifying dispersion in single-cell data to enable more robust quantification and study of transcriptional variability. We also measured transcriptional variability in three scRNA-seq datasets across two sequencing technologies and demonstrated that it can be used to identify candidate genes and pathways that can broaden our understanding of biological processes.

We first tested the sensitivity of the Gini index, Fano factor/VMR, CV, $CV^2$, variance, and Shannon entropy to variability in scRNA-seq data through simulations and found that overall Fano factor appears to be a reliable metric of dispersion in single-cell data among those tested. Despite its common use in the literature, entropy behaves less reliably in our simulations. The differences in the scale-invariance in entropy for different sampling distributions have been previously described for certain distributions. These likely result from differences in linear versus logarithmic scaling of a distribution, and how the closed-form function of entropy relates to its dependence on the first and second central moments of a distribution. We also found that the Gini index and metrics like the VMR have differing definitions of noisy, or variable, distributions. Additionally, change in the VMR appears to be relatively independent of a number of gene characteristics, including transcript level, gene length, promoter GC content, and gene evolutionary age.

Of note, our findings may also have implications for feature selection approaches in scRNA-seq analysis. Several existing methods developed for feature selection are based on VMR/Fano factor, and this step in turn affects downstream analysis tasks such as cell type identification and clustering [41,42]. Previous work has shown that highly variable genes influence the quality of scRNA-seq data integration, and our results imply that VMR would likely be a more robust choice for selecting these features in future projects, particularly those aimed at data integration across large cell atlas projects. VMR has also been used successfully to assess gene-level variability with biological implications in the context of innate immunity [43,44]. We anticipate that variable genes are context-dependent, and the magnitude of dispersion will likely be different across scenarios.

One key challenge of studying transcriptional variability in scRNA-seq data is that it can be difficult to deconvolve biological and technical noise. The latter can be introduced at many points in the scRNA-seq protocol such as during cell separation and capture, lysis, reverse transcription of mRNA, and amplification of cDNA. As a result, scRNA-seq data can have high technical noise within and between replicates. Technical noise is currently addressed through a variety of methods. During sequencing, the use of unique molecular identifiers (UMIs) reduces technical noise in scRNA-sequencing. Technical noise is also reduced by preprocessing steps in scRNA-seq analysis, such as filtering, scaling, and normalization [45]. In addition to these preprocessing steps, we excluded genes with low average expression from our transcriptional variability analysis to further minimize the influence of technical noise. Our current work focused on measuring transcriptional variability, and the simulations emulated biological noise without including an additional parameter for technical noise.

We demonstrated the utility of measuring transcriptional variability in independent scRNA-seq studies focused on the impact of matHG and T21 on mouse heart physiology and specifically in cardiomyocytes, as well as developmental time points during mouse neurogenesis. We found that genes with the largest changes in transcriptional variability were largely independent of DEGs. After performing GSEA and TF motif enrichment using DEGs and separately using genes with

largest absolute change in VMR, we found that genes with high transcriptional variability identified new pathways and TFs that have also previously been linked to associated disease phenotypes.

For example, our KEGG pathway and TF motif enrichment analyses of genes with large changes in transcriptional variability revealed components of AP-1 regulation (e.g., *c-Fos* and *Jun*). It is known that c-Fos is upregulated in cardiomyocytes cultured in high glucose [46] and in cardiomyocytes of diabetic rats [47]. matHG increases the amount of reactive oxygen species (ROS) in cardiomyocytes, and it has been suggested that the increase in ROS activates NF-kB, which in turn upregulates c-Fos [47]. These results increase our confidence in the ability of transcriptional variability to identify biologically relevant information. To this end, our analysis also revealed new potential targets for future research on the mechanisms that contribute to the hyperglycemic response, and more generally, cardiomyopathy. One example is *Tead4*, a TF in the Hippo pathway whose motif was enriched among the 100 genes with the largest absolute change in VMR at E9.5. The Hippo pathway functions through TEAD and AP-1 axes and was enriched in GSEA at E9.5 among the 1000 genes with largest change in VMR.

In conclusion, this work demonstrates that statistical modeling of transcriptional variability can be applied to identify pathways and genes beyond what is observed from differential expression analysis. Our results highlight the value of including computational analysis of transcriptional variability in single-cell data. Future work can be aimed at developing principled statistical tests to evaluate differences in VMR between conditions or datasets to further extend the applicability of modeling noise in scRNA-seq data.

## Methods

### Generating simulated scRNA-seq data

The sampling distributions for our simulations were chosen for their ability to model single-cell RNA-seq counts. For example, the Poisson($\lambda$) distribution models the probability that k UMIs align at a given locus, assuming UMI counts follow a Poisson process. Distributions like the negative binomial and beta-Poisson can be thought of as a mixture of a measurement model and an observation model, where the Poisson models the reads captured by scRNA-sequencing, and the gamma and beta distributions respectively model the true expression of reads. For all distributions except the beta-Poisson, the simulation assumes that the transcription levels of all genes and all cells in a matrix come from the same distribution. While this assumption does not reflect biology, the simplified framework allows for the study of metric performance and behavior. For each distribution, only the dispersion parameter was modified. Within each family of distributions, the relative change in the theoretical variance between instances of the distribution was fixed to allow for better assessment of the relative change in the metrics as the dispersion in the underlying sampling distribution increased.

The parameters of the sampling distribution were chosen to simulate real scRNA-seq counts. The mean of scRNA-seq counts is often less than 1 but can also range up to 10 [48]. The median is often 0, and the range of the counts can include values in the hundreds. Parameters were chosen to approximate these properties while also spanning a range of dispersion. For the Poisson distribution, we varied the lambda parameter $\lambda$ from 0.2 to 2278, where $\lambda$ increased exponentially. For the negative binomial distribution, the success probability $p$ was set to 0.5, and the dispersion parameter $r$ varied from 0.6 to 10, increasing exponentially. The dispersion parameter of the negative binomial distribution is the shape parameter of its gamma distribution. For the beta-Poisson distribution, the four-parameter beta-Poisson from the BPSC package [28] was used, where $\lambda_1 \sim$ Uniform(0, 1000), $\lambda_2 \sim$ Uniform(0, 1), $a \sim$ Uniform(0, 1), scale factor $c$ increased exponentially from 0.67 to 100, and $\beta \sim c$ Uniform(0, 1). For the uniform distribution, the lower limit of the distribution was set to 0 and the upper limit increased exponentially from 1 to 107.

For each family of distributions, we also varied the size of the simulated dataset. Each instance of a distribution generated 50 matrices of simulated counts with 500 cells and $m$ genes, where $m$ ranged from 1000 to 10800. Each instance of a distribution also generated 50 matrices of simulated counts with $n$ cells and 5000 genes, where $n$ ranged from 500 to 5400. This setup allowed us to study concurrently the effects of theoretical variance and dataset size on dispersion metrics.

 

## Modeling overdispersed counts and zero inflation

We modeled scenarios where the UMI counts may be overdispersed using the Poisson-lognormal model. Specifically, counts matrices were simulated using the R package poilog. The matrices ranged from 1000 to 4800 genes, and all had 500 cells. For each sample size, 100 matrices were simulated from instances of the Poisson lognormal distribution with varying amounts of dispersion. Six metrics were applied to each count matrix to quantify noise: variance, VMR, Gini index, Shannon entropy, CV, and $CV^2$.

To simulate zero inflated counts, we utilized a hurdle negative binomial distribution. We tested two probabilities of seeing a non-zero count: 0.1 (low) and 0.9 (high). For each probability of a non-zero count, or hurdle parameter, the dispersion in the negative binomial component of the distribution was varied via the size parameter. One hundred size parameters were tested, ranging from 0.06 to 10. The theoretical variance between each instance of the hurdle negative binomial increased at a fixed exponential rate. For each instance of the hurdle negative binomial (i.e., a given hurdle and size parameter), fifty counts matrices were simulated with varying numbers of cells. The smallest counts matrices contained 50 cells, and then largest counts matrices contained 5400 cells.

To evaluate the performance of each statistical measure of noise across the simulated counts matrix, the relative change in each metric as dispersion increased was calculated.

## Gene-gene correlations

To further add complexity to the simulations and specifically address the scenario that UMI counts across cells might not follow the assumption of being independent and identically distributed, we also simulated counts matrices with weak, medium, and strong gene-gene correlations for the Poisson and negative binomial distributions as the two most representative statistical distribution models for single cell data. Using the R package SimMultiCorrData, we simulated counts matrices with 100 genes each and with cell numbers ranging from 500 to 1500. For each sample size, we simulated experiments where genes had an average correlation of 0.1 (weak), 0.4 (medium), and 0.7 (strong).

For each of the simulated matrices, we again estimated the noise of each gene using six metrics: variance, VMR, Gini, Shannon entropy, CV, and $CV^2$. The mean of these metrics across each count matrix was then normalized relative to means from count matrices generated with varying amounts of dispersion to assess the sensitivity of the metrics to changing dispersion. It is important to note that measuring the relative change in CV values carries limitations as the CV (dividing the standard deviation by the mean) is not a linear variable. The comparison and interpretation of CV values of distributions with similar standard deviations but shifted means is therefore more challenging, particularly with distributions such as the Poisson, where mean and variance are inherently connected.

## Normalizing change in dispersion metrics

To normalize the change in the metrics as the dispersion in the simulated counts increased, we calculated the relative and ratio change. For all instances of a distribution with the same number of cells and same number of genes, we calculated the median of each metric across the simulated counts matrix. In other words, let $A_1$, $A_2$,…, $A_n$ be matrices of the same size and generated from the same distribution family. $A_1$ is generated from an instance of the distribution with lower dispersion than $A_2$, and so on. The medians of each metric across these matrices are $m_1$, $m_2$,…, $m_n$. The relative change of each metric was calculated as $\frac{m_2-m_1}{m_1}$, $\frac{m_3-m_2}{m_2}$,…, $\frac{m_n-m_{n-1}}{m_{n-1}}$. We also calculated the ratio change of each metric as $\frac{m_2}{m_1}$, $\frac{m_3}{m_2}$,…, $\frac{m_n}{m_{n-1}}$. The two normalizations produced equivalent heatmaps, and the ratio change heatmaps can be found in S1- S4 Figs.

As the increase in theoretical dispersion was fixed, one would expect the normalized relative change values to similarly be fixed. To better quantify this property, we calculated the mean relative change for each metric for each simulation. The deviation of individual relative change values from the mean relative change was calculated. To quantify the spread of the relative change values, the percentage of relative change values within 0.01, 0.005, and 0.0025 of the mean relative change was calculated to approximate the spread of the relative change values.

## Single-cell RNA-seq datasets

UMI counts from Manivannan et al. (2022) were obtained from the Gene Expression Omnibus (GSE193746). In Manivannan et al., the authors performed scRNA-sequencing using a 10x Genomics Chromium controller on the cardiogenic region of embryonic mouse hearts at E9.5 and E11.5. Their experiment included control embryos and matHG embryos, with ≥ 6 samples per time point per condition. matHG embryos are embryos harvested from mothers with blood glucose ≥ 200 mg/dl which was chemically induced via Streptozotocin administration. Reads were aligned to GRCm38.

Using Seurat v5.0.1, cells with fewer than 500 or greater than 7000 genes, fewer than 800 UMIs, and with more than 20% mitochondrial genes were filtered out. Genes that were expressed in fewer than 10 cells were removed. Clustering was performed on the remaining data, and clusters expressing *Myh7, Ryr2, Ttn, Mybpc3, Actn2, Tnnc1, Actc1, Myh6, Tnnt2, Acta2, Acta1*, and *Tnnc3* were identified as cardiomyocytes. We identified 3502 of 12951 cells as cardiomyocytes, with 760 of those belonging to the control E9.5 condition, 827 to the matHG E9.5 condition, 752 to the control E11.5 condition, and 1163 to the matHG E11.5 condition. After isolating the cardiomyocytes, we filtered out genes with mean normalized, scaled expression < 0.05 to minimize the impact of technical noise on transcriptional variability in downstream analyses. We also removed mitochondrial and ribosomal genes. The VMR was calculated for each gene on the normalized counts per time point per condition, and we then took the difference in the VMR between the control and matHG conditions at E9.5 and at E11.5 for each gene. Genes were ranked by their absolute change in VMR between control and matHG conditions. DEGs were identified using the default Wilcoxon Rank Sum test in Seurat. All analyses were performed using R v4.3.2.

The same methods were applied to UMI counts from Lana-Elola et al. (2024) (GSE196447). In Lana-Elola et al., the authors performed scRNA-sequencing using a 10x Genomics Chromium controller on E13.5 mouse hearts. Two WT samples and one Dp1Tyb sample were sequenced and aligned to GRCm38. Dp1Tyb mice are a mouse model for T21 that have an additional copy of the Hsa21 orthologous region on Mmu16. In this dataset, we identified 830 out of 1713 cells as cardiomyocytes across the control and T21 conditions, with 639 of those cells belonging to the control condition and 191 in the T21 condition.

UMI counts from Yuzwa et al. (2017) (GSE107122) were also similarly analyzed. The authors performed scRNA-sequencing using Drop-seq, a droplet based single cell capture method that utilizes 3' end-counting technique [39]. Brains from 12 E11.5 embryos, 16 E13.5 embryos, 13 E15.5 embryos, and 8 E17.5 embryos were analyzed and aligned to mm10. After clustering, neurons were identified using the markers *Tubb3, Tbr1, Foxp2, Satb2, Bhlhe22, Sema6d,* and *Unc5d*, as mentioned in Yuzwa et al.

## Gene set enrichment analyses and inference of regulatory motifs

GSEA was performed using KEGG pathways using the R package clusterProfiler v4.10.1. GSEA was performed in two ways. First, the 1000 most significant DEGs were ranked using their average log-fold change. Second, the 1000 genes with the largest absolute change in VMR were ranked using their signed change in VMR. The two ranked gene lists were then separately analyzed using GSEA. For KEGG GSEA, Benjamini-Hochberg method was used for multiple testing correction, and the adjusted p-value cutoff was set at 0.05. DEGs were ranked by adjusted p-values and had an FDR of ≤ 20%.

TF motif enrichment analysis was performed using the R package RcisTarget v1.23.1. We looked at motifs enriched among the 100 most significant DEGs, and among the 100 genes with largest absolute change in VMR. For each gene, motif annotations within 10 kbp of the gene were considered. Enriched motifs were used to predict TFs and their associated genes. Enriched TFs and their target genes were then filtered for expression in the cardiomyocyte dataset.

For KEGG GSEA, the background gene set was all genes in the mouse genome from org.Mm.eg.db v3.18, which is based on Entrez Gene identifiers. For RcisTarget, the background for motif enrichment was a database of gene-motif rankings generated from GRCm38 (mm10) that included 10 kbp upstream and downstream of each gene [37]. Motif enrichment analysis returns normalized enrichment score (NES).

**Testing for correlation between change in transcriptional variability and gene characteristics**

Kendall's rank correlation coefficient was used to measure all correlations. For all comparisons, only genes with mean scaled and normalized expression > 0.05 were considered to minimize the impact of technical noise on quantifying transcriptional variability. Additionally, all comparisons are performed using the absolute value of the change in VMR between control and disease conditions. Here, transcript level was defined as the mean normalized counts of a gene across both the control and the disease conditions at a given time point, if applicable. This is because transcriptional variability is calculated on the normalized counts. Promoter sequences were defined as sequences within 1000 bp of the transcription start site and were obtained using the UCSC mm10 genome and annotation database. Evolutionary age was assessed using phylostrata from a phylomap generated by Barrera-Redondo et al. using GenEra [49].

## Supporting information

**S1 Fig. Relative sensitivity of dispersion metrics to increasing transcriptional variability as modeled by the Poisson distribution.** Heatmaps of each metric applied to simulated counts drawn from instances of the Poisson distribution. The dispersion in Poisson (as determined by $\lambda$) increases across the y-axis. The size of the simulated data increases across the x-axis, as determined by the number of genes (A) or by the number of cells (B) in each counts matrix.
(TIF)

**S2 Fig. Relative sensitivity of dispersion metrics to increasing transcriptional variability as modeled by the negative binomial distribution.** Heatmaps of each metric applied to simulated counts drawn from instances of the negative binomial distribution. The dispersion in the negative binomial (as determined by size parameter $r$) increases across the y-axis. The size of the simulated data increases across the x-axis, as determined by the number of genes (A) or the by number of cells (B) in each counts matrix.
(TIF)

**S3 Fig. Relative sensitivity of dispersion metrics to increasing transcriptional variability as modeled by the beta-Poisson distribution.** Heatmaps of each metric applied to simulated counts drawn from instances of the beta-Poisson distribution. The dispersion in the beta-Poisson distribution increases across the y-axis, as determined by a scale factor of shape parameter $\beta$. The size of the simulated data increases across the x-axis, as determined by the number of genes (A) or by the number of cells (B) in each counts matrix.
(TIF)

**S4 Fig. Relative sensitivity of dispersion metrics to increasing transcriptional variability as modeled by the uniform distribution.** Heatmaps of each metric applied to simulated counts drawn from instances of the uniform distribution. The dispersion in the uniform distribution increases across the y-axis, as determined by maximum value $b$. The size of the simulated data increases across the x-axis, as determined by the number of genes (A) or by the number of cells (B) in each counts matrix.
(TIF)

**S5 Fig. Performance of dispersion metrics remains consistent in the presence of simulated gene-gene correlation.** (A) Heatmaps of each metric applied to simulated counts drawn from instances of the negative binomial distribution with weak gene-gene correlation. The dispersion in the sampling distributions increases across the y-axis, as determined by the sigma parameter. The size of the simulated data increases across the x-axis, as determined by the number of genes in each counts matrix. (B) Heatmaps of relative change in each metric applied to counts from (A). (C) Heatmaps of each metric applied to simulated counts drawn from instances of the negative binomial distribution with strong gene-gene correlation. (B) Heatmaps of relative change in each metric as applied to counts from (C). (E, F) Histograms of deviation of relative change from mean relative change for each metric as calculated from the distributions described in (A, B).
(TIF)

**S6 Fig. VMR, Variance, CV, and CV² are robust metrics of dispersion on simulated Poisson-lognormal counts.** (A) Heatmaps of each metric applied to simulated counts drawn from instances of the Poisson-lognormal distribution. The dispersion in the sampling distributions increases across the y-axis, as determined by the sigma parameter. The size of the simulated data increases across the x-axis, as determined by the number of genes in each counts matrix. (B) Heatmaps of relative change in each metric applied to counts from (A). (C) Histograms of deviation of relative change from mean relative change for each metric as calculated from the distributions described in (A).
(TIF)

**S7 Fig. Relative sensitivity of dispersion metrics to increasing transcriptional variability as modeled by the hurdle negative binomial distribution.** Heatmaps of each metric applied to simulated counts drawn from instances of the hurdle negative binomial distribution with hurdle parameters, which is the probability of a non-zero count, of 0.1 (A) and of 0.9 (D). The dispersion in the sampling distributions increases across the y-axis, as determined by the size parameter. The size of the simulated data increases across the x-axis, as determined by the number of genes in each counts matrix. (B, E) Heatmaps of relative change in each metric applied to counts from (A, D). (C, F) Histograms of deviation of relative change from mean relative change for each metric as calculated from the distributions described in (A, D).
(TIF)

**S8 Fig. Change in transcriptional variability is not correlated with gene characteristics in a T21 mouse dataset.** Correlation of absolute change in VMR between control and T21 conditions with (A) mean normalized expression, (B) gene length, (C) promoter GC content, and (D) gene phylostrata in the T21 dataset from Lana-Elola et al. (2024). Kendall's rank correlation coefficient is shown for each comparison.
(TIF)

**S9 Fig. Tgfβ and Hif1a signaling pathways contain a mixture of DEGs and non-differentially expressed genes in a matHG dataset.** Scatterplots of genes in the Tgfβ and Hif1a signaling pathways plotted by their change in VMR and average log-fold change in the matHG dataset from Manivannan et al. (2022). The Tgfβ signaling pathway was enriched in GSEA of the 1000 genes with largest change in VMR at E9.5. The average log-fold change shown for genes in the Tgfβ signaling pathway is derived from the data at E9.5. *Hif1a* was enriched in the TF motif enrichment analysis of the 100 genes with largest change in VMR at E11.5. The average log-fold change shown for genes in the Hif1a signaling pathway is derived from the data at E11.5.
(TIF)

**S10 Fig. Transcriptional variability identifies pathways involved in neuronal development.** (A) Venn diagram of the 1481 significant DEGs and the 1481 genes with largest absolute change in VMR between E11.5 and at E17.5, including 470 and 711 neurons at E11.5 and E17.5, respectively. Significant DEGs are genes that had an adjusted p-value < 0.05. (B) GSEA using KEGG pathways on the 1000 most significant DEGs, ranked by p-value, and on the 1000 genes with largest absolute change in VMR, ranked by change in VMR. (C) TF motif enrichment was performed on the 100 most significant DEGs and on the 100 genes with largest absolute change in VMR, and the top 5 enriched motifs from each are shown [37]. (D) Dendrogram of TFs enriched among the 100 genes with largest absolute change in VMR and their target genes. Opacity of edge weights is determined by the change in VMR of the genes. DEGs are shown in gray.
(TIF)

**S1 Table. Transcriptional variability analysis, GSEA, and TF motif enrichment analysis of a matHG dataset.** Sheets 1–2: Change in VMR between control and matHG conditions for each gene included in analysis at E9.5 (Sheet 1) and at E11.5 (Sheet 2). Sheets 3–4: DEGs between control and matHG conditions at E9.5 (Sheet 3) and at E11.5 (Sheet 4). Sheet 5: Top 100 DEGs by absolute fold change and top 100 genes with largest absolute change in VMR, as well as the genes that are contained in both lists, across two timepoints. All comparisons are between the control and matHG conditions. Sheet 6: KEGG

pathways resulting from GSEA using the 1000 most significant DEGs, ranked by log-fold change versus from GSEA using the top 1000 genes with largest absolute change in VMR, ranked by change in VMR. Sheets 7–8: Enriched motifs at E9.5 (Sheet 7) and at E11.5 (Sheet 8) among the 100 most significant DEGs, and among the top 100 genes with largest absolute change in VMR. Each motif has a normalized enrichment score (NES), AUC score, predicted TFs, number of enriched genes, and enriched genes. Sheets 9–10: Predicted enriched TFs from the motifs found in Sheets 7–8, after filtering out TFs that are not expressed in the data at their respective time points. Enriched TFs that are only enriched among the top 100 genes with largest absolute change in VMR, and not among the 100 most significant DEGs, are also listed for each time point. (XLSX)

**S2 Table. Transcriptional variability analysis, GSEA, and TF motif enrichment analysis of a T21 dataset.** Sheet 1: Change in VMR for each gene between control and T21 mouse model (Dp1Tyb) conditions. Sheet 2: DEGs between control and T21 conditions. Sheet 3: Top 100 DEGs by absolute fold change and top 100 genes with largest absolute change in VMR, as well as the genes that are contained in both lists. All comparisons are between the control and T21 mouse model (Dp1Tyb) conditions. Sheet 4: KEGG pathways resulting from GSEA using the 1000 most significant DEGs, ranked by log-fold change versus from GSEA using the top 1000 genes with largest absolute change in VMR, ranked by change in VMR. Sheet 5: Enriched motifs among the 100 most significant DEGs, and among the top 100 genes with largest absolute change in VMR. Each motif has a normalized enrichment score (NES), AUC score, predicted TFs, number of enriched genes, and enriched genes. Sheet 6: Predicted enriched TFs from the motifs found in Sheet 5, after filtering out TFs that are not expressed in neither the control nor T21 conditions. Enriched TFs that are only enriched among the top 100 genes with largest absolute change in VMR, and not among the 100 most significant DEGs, are also listed. (XLSX)

## Acknowledgments

We thank John H. Day in the Boyer lab for helpful discussions.

## Author contributions

**Conceptualization:** Laurie A. Boyer.

**Formal analysis:** Tina Chen.

**Investigation:** Tina Chen.

**Project administration:** Laurie A. Boyer.

**Supervision:** Divyansh Agarwal.

**Writing – original draft:** Tina Chen.

**Writing – review & editing:** Laurie A. Boyer, Divyansh Agarwal.

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
