## [Decision Letter · Decision Letter 0]

20 Aug 2025

PCOMPBIOL-D-25-01049

Assessment of dispersion metrics for estimating single-cell transcriptional variability

PLOS Computational Biology

Dear Dr. Chen,

Thank you for submitting your manuscript to PLOS Computational Biology. After careful consideration, we feel that it has merit but does not fully meet PLOS Computational Biology's publication criteria as it currently stands. Therefore, we invite you to submit a revised version of the manuscript that addresses the points raised during the review process.

Please submit your revised manuscript within 60 days Oct 20 2025 11:59PM. If you will need more time than this to complete your revisions, please reply to this message or contact the journal office at ploscompbiol@plos.org. Please include the following items when submitting your revised manuscript:

We look forward to receiving your revised manuscript.

Kind regards,

Zhixiang Lin

Academic Editor

PLOS Computational Biology

Stacey Finley

Section Editor

PLOS Computational Biology

**Additional Editor Comments :**

The reviewers raise many critical points that must be addressed in a substantially revised manuscript.

**Journal Requirements:**

At this stage, the following Authors/Authors require contributions: Tina Chen, Divyansh Agarwal, and Laurie A. Boyer. Please ensure that the full contributions of each author are acknowledged in the "Add/Edit/Remove Authors" section of our submission form.

**Reviewers' comments:**

Reviewer's Responses to Questions

**Comments to the Authors:**

**Please note that one of the reviews is uploaded as an attachment.**

Reviewer #1: Please see the attached review, recommending acceptance after major revisions.

Reviewer #2: The manuscript by Chen et al. addresses methods for quantifying variability in scRNA-seq data. Given the rapid expansion of single-cell approaches, this is a timely and important topic, especially considering the wide array of variability metrics currently in use.

The authors compare several measures: Gini index, variance-to-mean ratio (VMR), variance, and Shannon entropy, using simulated scRNA-seq datasets. They show that VMR exhibits particularly favorable properties, including linear scaling with dispersion (Figs. 1–3). This analysis is conducted to a high standard; however, the authors do not include other commonly used metrics, such as the coefficient of variation (CV) [or CV^2]. I believe the data simulations and analytical framework would benefit from a more formal description in the text.

I have concerns regarding the second part of the analysis, which uses previously published data. In particular, the authors use a developmental dataset to explore relationships between VMR changes and sequence features (Fig. 4 and similar analysis in Fig. S6 for analysis of T21). While they report limited correlations, I wonder whether this is because VMR changes between the two conditions are, in fact, minimal. Prior studies have shown that VMR remains relatively stable across conditions for individual genes, at least in the context of the innate immune response (see Bagnall et al., 2020 [https://doi.org/10.1016/j.cels.2020.08.007] and Alachkar et al., 2023 [https://doi.org/10.3389/fmolb.2023.1176107]). See specific comments below.

In Figure 5, the authors demonstrate that differentially expressed genes (DEGs) and high-VMR genes are largely non-overlapping in the developmental dataset. This observation supports the idea that basally expressed genes, that is those not induced in a given context, can nonetheless be highly variable. The authors suggest that VMR analysis could provide novel insights beyond standard DEG approaches. However, they fall short of demonstrating specific advantages in the context of the data analyzed. Notably, the identification of highly variable genes (HVGs) is already a standard and essential step in 10x Genomics scRNA-seq workflows, including in tools like Seurat (R), Scanpy (Python), Cell Ranger, and Bioconductor packages such as scran and scater.

I encourage the authors to include additional, more complex datasets encompassing multiple conditions. This would allow for a more rigorous assessment of the utility of VMR-based analysis and strengthen the broader applicability of their conclusions.

The manuscript is generally well written, but some sections would benefit from further clarification- please see specific comments below.

Major comments:

1)

Fig 1: line 400- Some aspects of the data simulations are not clear. The numerical scheme should be described more formally. The authors state for one of the distributions: “For the Poisson distribution, we varied the lambda parameter λ from 0.2 to 2278, where λ increased exponentially.” Similar scheme is described for other distributions.

There is varying number of cells and genes per individual cells in simulation. The authors must simulate the range of expression between different genes, i.e., low expressing vs. high expressing. It seems that each gene per individual cell is sampled independently from the reference distribution. The number of different levels of lambda (e.g., for Poisson) seems to be fixed to 10 for all distributions. Should simulation of larger number of genes require a finer gradation of the lambda parameter (and corresponding parameters for other sampling distributions)? After all 10 seems to be quite low comparing to number of genes samples (up to 10,000). The simulation scheme should be also represented in more details in the Figure 1.

2)

Fig. 2. Authors should provide some quantification (beyond a simple drawing of the heat map) as they claim that VMR is most sensitive. Gini appear to exhibit similar to VMR linear scaling of sensitivity, although as author describe with an inverse relationship between index and dispersion. What about other measures, in particular CV (or CV^2) typically used for the quantification of noise- I presume the scaling will not be linear.

3)

Figure 4 presents an analysis that I find somewhat oversimplified. The relationship between gene expression variability and sequence features is a complex topic that has already been explored in prior literature, particularly in an evolutionary context. Notably, Hagai et al. (2018) [https://doi.org/10.1038/s41586-018-0657-2] demonstrated that variability across species and contexts is linked to features such as promoter architecture and coding sequence properties. These findings highlight that gene regulatory architecture and evolutionary pressures can shape the intrinsic variability of gene expression.

Authors report lack of correlation between VMR and features. However, I find that the changes in VMR are overall very small, which might contribute to the lack of statistical power.

Given a fold change (FF) in VMR defined as (based on Table S1):

FF = abs (change in VMR)/max(VMR control, VMR MatHG)

Out of 881 genes (E9.5 change in VMR), there is only ~10% (950) of genes exhibits FF greater than 0.2 (20%), and 2% (184) genes greater than 0.3.

For the E11.5, there is 7% of genes with FF>0.3, and 35% with FF>0.2.

Therefore, the actual changes of VMR are very small overall. This is also the case of the second dataset analysed by authors (Fig. S6).

I refer the authors to work by Bagnall et al and Alachkar et al, who showed that VMR remains invariant across different environmental conditions, while the relative changes of variance and mean in mRNA/read counts can be related to parameters of transcriptional bursting. This work suggests that no changes in VMR across conditions are expected, which is in a broad agreement with what authors are finding.

The dataset presented by authors includes only two-time-points (as far as I understand), but perhaps it is worth to test if the differences between VMRs are statistically significant in the dataset present. Perhaps analysis of the data grouped according to range of VMR values (similar to analyses performed by Hagai et al.) will show stronger trends. I would also encourage authors to look at other datasets which include more experimental conditions to extend this analysis beyond 2 time-points. The authors' approach in Figure 4 could be strengthened by incorporating a more nuanced interpretation of sequence-feature relationships.

4)

In Figure 5, the authors essentially argue that basally (or constitutively) expressed genes in a given context can be highly variable, often more variable than differentially expressed genes (DEGs). While they show that the sets of high-VMR genes and DEGs are largely non-overlapping, this observation is not novel. It is well established that basally expressed genes can display substantial variability across single cells. Moreover, the finding that DEGs are less variable is expected: these genes often respond to strong, uniform signals (e.g., cytokines or developmental cues), resulting in synchronized transcriptional responses across cells. In contrast, variability in basal gene expression may reflect asynchronous states, local microenvironmental influences, or stochastic differences in transcription factor activity, all of which can lead to high cell-to-cell variability, even in the absence of external perturbation.

While this finding confirms existing knowledge, the authors’ analysis lacks deeper functional insight. I would strongly recommend including more complex or richer datasets, ideally encompassing multiple conditions or cell types to more convincingly demonstrate the added value of VMR-based analysis. Simply showing that VMR and DEG analyses yield different sets of genes is not sufficient. A clearer demonstration of the functional relevance of high-VMR genes would strengthen the manuscript and better support the claim that VMR captures biologically meaningful variability beyond standard DEG approaches.

Minor comments:

31: “the Gini index displayed paradoxical behavior” please spell out what you mean here – the abstract should capture some details of the main findings

57: “Single-cell RNA sequencing (scRNA-seq) approaches have enabled the study of gene expression at high resolution. “ So does bulk-cell RNA-seq, be more specific.

127: ”we simulated counts matrices from 100 instances with differing levels of dispersion” Please clarify instances as described in the Methods section? This phrase also appears in the Methods (line 397).

137/139: “Schema of/for” should be “Schematic of…”

388 : Methods: in places unclear, for example: “For example, the biological intuition behind Poisson(λ = k) is that it is the probability k UMIs align at a given loci assuming a Poisson distribution.” should be “For example, the Poisson(λ) distribution models the probability that k UMIs align at a given locus, assuming UMI counts follow a Poisson process.”

Fig S1-4 captions: I am guessing that the numbers 0.573 to 83.8 correspond to Poisson parameters (equivalent of the variance) across genes (in the Raw data on the left panel). Corresponding numbers for the normalised data appear to be substantially different, especially in the case of Poisson, please explain how are they calculated?

Figs S1 and S3 seem to be identical.

144: Why the differences in the scale-invariance in Entropy for different sampling distributions, please provide some insights.

153: Figs S1, S2 and S4 are referred out of order in the text (i.e., after the S3 and S5).

176: The Figure 3A refers to negative binomial, yet the discussion following the initial stamen is related to Poisson: “An intuition for this paradoxical behavior can be observed from the shapes of assessed distributions (Fig 3A). For example, in the Poisson distribution, the Gini index indicates highest noise when the mean and variance of the distribution are less than 1, which corresponds to data with a high proportion of zeros. When the mean and variance of the Poisson are greater than 1…” Of note, mean and variance of Poisson is always equal to lambda, please make sure that you clarify the difference between the sampling distribution and the distribution obtained in the simulated data.

190-199- It would be useful to use a more relevant intuitive example, perhaps referring to read counts.

Reviewer #3: The paper systematically compares four dispersion metrics for scRNA-seq—Gini index, variance-to-mean ratio (VMR), variance, and Shannon entropy—via simulations across several distributions (Poisson, negative binomial, beta-Poisson, uniform), then applies the preferred metric to murine cardiomyocyte datasets (maternal hyperglycemia at E9.5/E11.5; and a trisomy-21 model). The study is well-structured and the analyses are compelling. however, minor revisions are needed before acceptance.

- The manuscript uses “scale-invariant” to mean independence from dataset size, which is well supported; however, VMR = variance/mean can still inherit a mean-dependence for many count distributions (e.g., NB), so the text should clarify what kind of scale-invariance is claimed (dataset size vs. expression-level scaling) and delineate conditions where VMR may still drift with mean.

- Methods indicate VMR was computed on normalized, scaled counts per time point/condition. This can distort count-level mean–variance relationships and complicate interpretation. Please recompute VMR on raw UMI counts after size-factor normalization (or Pearson residuals) without per-gene scaling; and show concordance between “raw-VMR” and “scaled-VMR” rankings; and (iii) discuss implications.

- You note that correlations between VMR change and gene features depend on the threshold used to filter lowly expressed genes. Include a sensitivity analysis (multiple thresholds) and report stability metrics for top VMR-change genes and enriched pathways.

-Simulations span Poisson/NB/beta-Poisson/uniform, but common scRNA-seq nuisances (zero inflation/hurdle behavior, ambient RNA, library-size heterogeneity) could be reflected more explicitly. Consider adding simulations with spike-in technical noise or a hurdle-NB to test robustness of metric rankings.

-Because VMR corresponds to the classic Fano factor in count data, please make that synonym explicit, and consider adding comparisons to CV² (variance/mean²) and to variance-stabilized residual approaches to underscore when VMR is preferable. The current comparison to “variance” and “entropy” is useful but could miss widely used dispersion baselines.

-You analyze matHG (E9.5/E11.5) and a T21 model. It would be helpful to quantify cross-timepoint and cross-dataset reproducibility of VMR-change rankings (e.g., rank-rank hypergeometric overlap, bootstrap stability). Also report how many cardiomyocytes per group underpin these estimates to contextualize variance in VMR itself.

-For the “top-k” overlaps (100/1000), specify how significance/ranking for DEGs and VMR-change sets were controlled (e.g., FDR for DEG list; whether any p-values or empirical nulls were used for VMR changes). Including q-value cutoffs and enrichment NES/FDRs will add rigor.

- For KEGG GSEA and RcisTarget, please describe background gene sets, species/annotation versions, and multiple-testing thresholds; given the non-overlap of KEGG pathways between DE and VMR lists, a permutation-based concordance test (e.g., gene-set co-enrichment) would be informative.

**Have the authors made all data and (if applicable) computational code underlying the findings in their manuscript fully available?**

Reviewer #1: Yes

Reviewer #2: Yes

Reviewer #3: Yes

PLOS authors have the option to publish the peer review history of their article (what does this mean? ). If published, this will include your full peer review and any attached files.

**Do you want your identity to be public for this peer review?** For information about this choice, including consent withdrawal, please see our Privacy Policy .

Reviewer #1: No

Reviewer #2: No

Reviewer #3: No

**Figure resubmission:**

**Reproducibility:**



---

## [Decision Letter · Decision Letter 1]

19 Jan 2026

PCOMPBIOL-D-25-01049R1

Assessment of dispersion metrics for estimating single-cell transcriptional variability

PLOS Computational Biology

Dear Dr. Chen,

Thank you for submitting your manuscript to PLOS Computational Biology. One of the reviewers still raises several comments regarding the manuscript. Therefore, we invite you to submit a revised version of the manuscript that addresses the points raised during the review process.

We look forward to receiving your revised manuscript.

Kind regards,

Zhixiang Lin

Academic Editor

PLOS Computational Biology

Stacey Finley

Section Editor

PLOS Computational Biology

**Additional Editor Comments:**

Please refer to the comments raised by the reviewer.

**Reviewers' comments:**

Reviewer's Responses to Questions

**Comments to the Authors:**

Reviewer #2: The authors have included additional theoretical and data analyses in the region, which overall have substantially improved the manuscript.

Re: The discussion of the extent of biological insight, specifically, the interpretation of the fact VMR and DEG analyses yield different sets of genes.

I suggest that the authors further examine the expression patterns of the genes showing changes in VMR.

In particular, it would be informative to determine whether the observed VMR changes reflect shifts in the fraction of expressing cells across the population, or instead correspond to 'analogue' changes in expression levels within individual cells. Such an analysis would align well with the authors’ discussion of this question and provide additional biological interpretation of the nature of VMR-associated genes (in comparison to DEGs).

Fig S10B, please provide consistent formatting for p-values (i.e., number of digits denoted)

Fig. 5C: “We found that enriched TF motifs differed between the two groups (Fig 5C, S1 Table). For example, distinct motifs found among the genes with largest change in VMR include Tead, Fos, Jun, and Ctcf.”

For clarity please label different motifs referred to in C (and presumably S10). What are the key motifs corresponding to DEGs as presented in the panel?

Reviewer #3: I have carefully reviewed the revised manuscript and the authors’ responses to my previous comments. All of my concerns have been adequately addressed, and the revisions have significantly improved both the quality and the technical rigor of the paper. The manuscript is now clear, well-structured, and methodologically sound.

In my opinion, the paper is suitable for publication.

**Have the authors made all data and (if applicable) computational code underlying the findings in their manuscript fully available?**

Reviewer #2: Yes

Reviewer #3: Yes

PLOS authors have the option to publish the peer review history of their article (what does this mean? ). If published, this will include your full peer review and any attached files.

**Do you want your identity to be public for this peer review?** For information about this choice, including consent withdrawal, please see our Privacy Policy .

Reviewer #2: No

Reviewer #3: No

**Figure resubmission:**
---

## [Decision Letter · Decision Letter 2]

16 Feb 2026

Dear Master's Student Chen,

We are pleased to inform you that your manuscript 'Assessment of dispersion metrics for estimating single-cell transcriptional variability' has been provisionally accepted for publication in PLOS Computational Biology.

Best regards,

Zhixiang Lin

Academic Editor

PLOS Computational Biology

Stacey Finley

Section Editor

PLOS Computational Biology

Reviewer's Responses to Questions

**Comments to the Authors:**

Reviewer #2: Authors satisfactorliy addressed my queries, I recommend this manuscript for publication.

**Have the authors made all data and (if applicable) computational code underlying the findings in their manuscript fully available?**

Reviewer #2: None

PLOS authors have the option to publish the peer review history of their article (what does this mean? ). If published, this will include your full peer review and any attached files.

**Do you want your identity to be public for this peer review?** For information about this choice, including consent withdrawal, please see our Privacy Policy .

Reviewer #2: No

---

## [Editor Report · Acceptance letter]

PCOMPBIOL-D-25-01049R2

Assessment of dispersion metrics for estimating single-cell transcriptional variability

Dear Dr Chen,

I am pleased to inform you that your manuscript has been formally accepted for publication in PLOS Computational Biology. Your manuscript is now with our production department and you will be notified of the publication date in due course.

With kind regards,

Anita Estes
